# Demographic Factors Predict Risk of Lymph Node Involvement in Patients with Endometrial Adenocarcinoma

**DOI:** 10.3390/biology12070982

**Published:** 2023-07-10

**Authors:** Eric M. Anderson, Michael Luu, Mitchell Kamrava

**Affiliations:** 1Department of Radiation Oncology, Cedars-Sinai Medical Center, Los Angeles, CA 90048, USA; mitchell.kamrava@cshs.org; 2Department of Biostatistics and Bioinformatics, Cedars-Sinai Medical Center, Los Angeles, CA 90048, USA; michael.luu@cshs.org

**Keywords:** endometrial adenocarcinoma, lymph nodes, age, race, lymphovascular invasion

## Abstract

**Simple Summary:**

The presence of lymph node involvement for patients with endometrial cancer guides treatment after surgery. Demographic factors in addition to pathologic tumor characteristics may more accurately predict the risk of lymph node involvement in this population. This study utilized a publicly available database of endometrial cancer patients diagnosed between 2004 and 2016. Pathologic primary tumor predictors of lymph node involvement were identified using statistical analysis. Among the 35,170 patients included, 2864 were node positive. Analysis revealed that younger patient age, black versus white race, increasing primary tumor stage and biologic aggressiveness, and tumor size were predictive of lymph node involvement. Both black versus white and other versus white race strongly predicted paraaortic lymph node involvement. Independent subset analyses of black and white women revealed that tumor grade was a stronger predictor of lymph node involvement among black women. In addition to standard pathologic tumor features, patient age and race are associated with a higher risk of regional lymph node involvement. This information may inform adjuvant treatment decisions and guide future studies.

**Abstract:**

The presence of lymph node positivity (LN+) guides adjuvant treatment for endometrial adenocarcinoma (EAC) patients, but recommendations regarding LN evaluation at the time of primary surgery remain variable. Sociodemographic factors in addition to pathologic tumor characteristics may more accurately predict risk of LN+ in EAC patients. Patients diagnosed between 2004 and 2016 with pathologic T1-T2 EAC who had at least one lymph node sampled at the time of surgery in the National Cancer Data Base were included. Pathologic primary tumor predictors of LN+ were identified using logistic regression. To predict overall, pelvic only, and paraaortic and/or pelvic LN+, nomograms were generated. Among the 35,170 EAC patients included, 2864 were node positive. Using multivariable analysis, younger patient age (OR 0.98, 95% CI 0.98–0.99, *p* < 0.001), black versus white race (OR 1.19, 95% CI 1.01–1.40, *p* = 0.04), increasing pathologic tumor stage and grade, increase in tumor size, and presence of lymphovascular invasion were predictive of regional LN+. Both black versus white (OR 1.64, 95% CI 1.27–2.09, *p* < 0.001) and other versus white race (OR 1.54, 95% CI 1.12–2.07, *p* = 0.006) strongly predicted paraaortic LN+ in the multivariable analysis. Independent subset analyses of black and white women revealed that tumor grade was a stronger predictor of LN+ among black women. In addition to standard pathologic tumor features, patient age and race were associated with a higher risk of regional LN+ generally and paraaortic LN+ specifically. This information may inform adjuvant treatment decisions and guide future studies.

## 1. Introduction

Endometrial cancer is typically locally confined but can involve regional lymph nodes. Clinically apparent early stage disease is initially managed with surgery with pathologic staging impacting adjuvant treatment decisions. The use of pelvic lymphadenectomy is controversial [1], and staging evaluation can include dissection of the pelvic lymph nodes with or without paraaortic sampling [2] or sentinel lymph node sampling [3,4]. A survival benefit was not seen with the use of pelvic lymphadenectomy in two randomized studies [5,6]. However, among patients found to have nodal involvement, systemic therapy and radiation have been shown to improve patient outcomes showing that pathologic nodal information is clinically meaningful [7,8].

Given these controversies, depending upon institutional practice patterns, the pathologic information available to determine nodal status could range from no lymph nodes evaluated to a full lymphadenectomy. Furthermore, a lack of pathologic nodal information could impact treatment recommendations for adjuvant therapy. Accurately estimating lymph node positivity risk from primary pathologic factors may help to personalize adjuvant treatment recommendations.

Pathologic risk factors of lymph node positivity (LN+) from the primary tumor, including lymphovascular invasion (LVI), grade, stage, and size have been identified in previous studies [9]. Furthermore, limited studies have evaluated potential patient sociodemographic factors as predictors of pathologic LN+. In this study, the National Cancer Data Base (NCDB) was queried with the goal of validating previously identified risk factors of nodal involvement in a large national patient sample, as well as assessing for additional predictors.

## 2. Materials and Methods

### 2.1. Patient Selection

The NCDB was queried, and 476,104 patients diagnosed in the years 2004–2016 with endometrial cancer were identified (Figure 1). Initial exclusion criteria were the absence of known pathologic information, including LVI, grade, or tumor size (*n* = 354,392). Patients were excluded if they had undifferentiated tumors. Patients without known nodal stage were excluded (*n* = 69,861). Patients were excluded for primary tumor stage other than T1a, T1b, or T2. Non-adenocarcinoma histology was an exclusion criterion (*n* = 9260).

### 2.2. Statistical Analysis

Multivariable imputation with fully conditional specification was used to impute missing values. Student *t* test, Mann–Whitney test (continuous variables), and the Pearson chi-square test (categorical variables) were used to compare baseline characteristics. Pathologic lymph node involvement (≥1 positive) was the primary end point of the study. Patients with American Joint Committee on Cancer (AJCC) N1 involvement corresponded to those with only pelvic lymph node positivity while AJCC N2 involvement corresponded to paraaortic +/− pelvic LN+. Tumor size was assessed postoperatively and defined by the evaluating pathologist.

Predictors of any regional LN+, only pelvic LN+, and paraaortic +/− pelvic LN+ were identified using logistic regression. The variable inflation factor was used to assess for multicollinearity [10]. Internal bootstrap resampling with 1000 replicates for model validation was used to generate an optimism corrected c-index [11]. Model data were used to generate nomograms to predict the probability of LN+. R statistical software (version 4.0.1; R Foundation, Vienna, Austria) was used, and statistical significance was defined at a level of 0.05 using a 2-sided test.

**Figure 1 biology-12-00982-f001:**
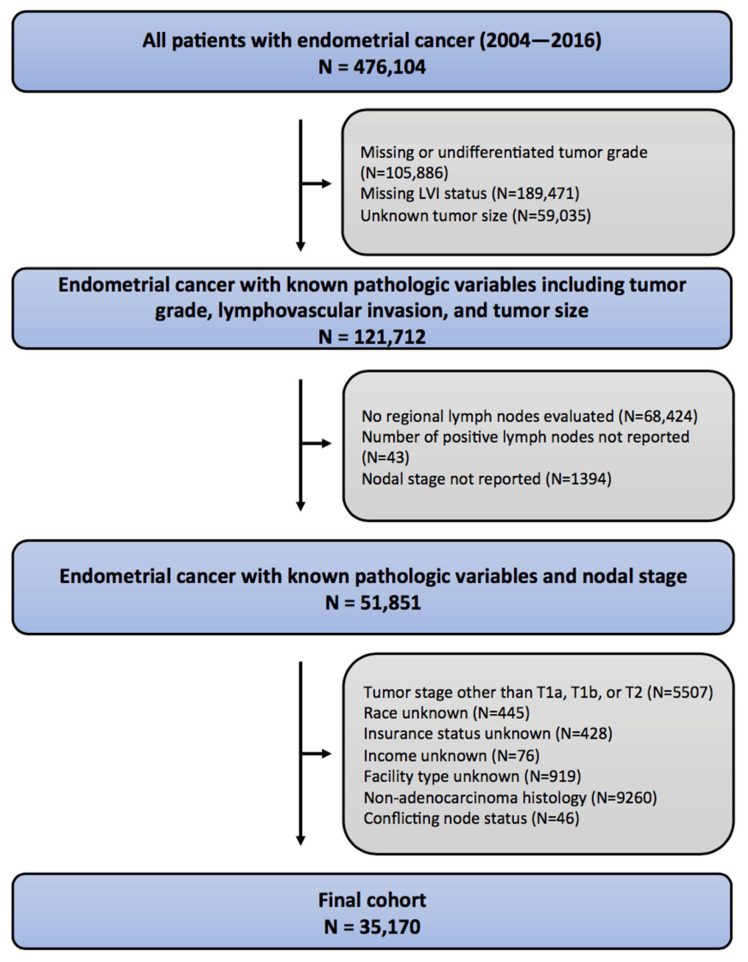
Consolidated standards of reporting trials (CONSORT) diagram describing inclusion/exclusion criteria for the study.

## 3. Results

There were 35,170 patients included in this study, and 2864 (8.1%) had a pathologically positive lymph node. The patient sociodemographic and tumor pathologic factors are listed in Table 1. Most patients in the study were white (31,274), while a smaller proportion were black (2329) or of another race (1567). Multiple factors were associated with LN+ on univariate analysis, including age (per 1 year increase, odds ratio [OR] 0.995, 95% confidence interval [CI] 0.991–0.999, *p* = 0.02), race (black versus white, OR 1.28, 95% CI 1.11–1.45, *p* < 0.001; other versus white, OR 1.12, 95% CI 0.94–1.34, *p* = 0.2), insurance status (private versus uninsured, OR 0.81, 95% CI 0.67–0.99, *p* = 0.04; Medicaid versus uninsured, OR 1.29, 95% CI 1.01–1.64, *p* = 0.04), higher income (OR 0.92, 95% CI 0.85–0.997, *p* = 0.04), more education (OR 0.92, 95% CI 0.85–0.99, *p* = 0.03), year of diagnosis (OR 1.03, 95% CI 1.007–1.05, *p* = 0.01), higher T stage (pT1b versus pT1a, OR 5.36, 95% CI 4.88–5.90, *p* < 0.001; pT2 versus pT1a, OR 10.66, 95% CI 9.52–11.94, *p* < 0.001), higher tumor grade (grade 2 versus grade 1, OR 2.22, 95% CI 2.02–2.44, *p* < 0.001; grade 3 versus grade 1, OR 3.48, 95% CI 3.13–3.86, *p* < 0.001), LVI positivity (OR 10.44, 95% CI 9.61–11.35, *p* < 0.001), and tumor size (OR 1.15, 95% CI 1.14–1.17, *p* < 0.001) (Table 2). Multivariate logistic regression analysis revealed multiple independent factors of LN+, including age (OR 0.98, 95% CI 0.98–0.99, *p* < 0.001), race (black versus white, OR 1.19, 95% CI 1.01–1.44, *p* = 0.04), insurance status (Medicaid versus uninsured, OR 1.42, 95% CI 1.09–1.88, *p* = 0.01), academic facility type (OR 1.11, 95% CI 0.995–1.22, *p* = 0.06), increasing primary tumor stage (pT1b versus pT1a, OR 3.08, 95% CI 2.78–3.42, *p* < 0.001; pT2 versus pT1a, OR 5.10, 95% CI 4.50–5.78, *p* < 0.001), higher tumor grade (grade 2 versus grade 1, OR 1.45, 95% CI 1.31–1.61, *p* < 0.001; grade 3 versus grade 1, OR 1.47, 95% CI 1.30–1.65, *p* < 0.001), presence of LVI (OR 6.44, 95% CI 5.88–7.05, *p* < 0.001), and tumor size (OR 1.0, 95% CI 1.04–1.06, *p* < 0.001) (Table 2).

There were multiple significant predictors of pelvic only LN+ including age (OR 0.98, 95% CI 0.98–0.99, *p* < 0.001), insurance status (private versus uninsured, OR 1.29, 95% CI 0.99–1.70, *p* = 0.06; Medicaid versus uninsured, OR 1.62, 95% CI 1.18–2.25, *p* = 0.003; Medicare versus uninsured, OR 1.35, 95% CI 1.02–1.81, *p* = 0.04), academic facility type (OR 1.13, 95% CI 1.002–1.27, *p* = 0.04), increasing primary tumor stage (pT1b versus pT1a, OR 2.91, 95% CI 2.58–3.28, *p* < 0.001; pT2 versus pT1a, OR 5.00, 95% CI 4.34–5.76, *p* < 0.001), higher tumor grade (grade 2 versus grade 1, OR 1.47, 95% CI 1.31–1.65, *p* < 0.001; grade 3 versus grade 1, OR 1.45, 95% CI 1.27–1.67, *p* < 0.001), presence of LVI (OR 5.77, 95% CI 5.20–6.39, *p* < 0.001), and tumor size (OR 1.04, 95% CI 1.03–1.06, *p* < 0.001) (Table 3).

The predictors of paraaortic +/− pelvic LN+ included age (OR 0.97, 95% CI 0.96–0.98, *p* < 0.001), race (black versus white, OR 1.64, 95% CI 1.27–2.09, *p* < 0.001; other versus white, OR 1.54, 95% CI 1.12–2.07, *p* = 0.006), increasing primary tumor stage (pT1b versus pT1a, OR 3.52, 95% CI 2.91–4.27, *p* < 0.001; pT2 versus pT1a, OR 5.33, 95% CI 4.28–6.65, *p* < 0.001), higher tumor grade (grade 2 versus grade 1, OR 1.38, 95% CI 1.15–1.66, *p* < 0.001; grade 3 versus grade 1, OR 1.40, 95% CI 1.14–1.72, *p* = 0.001), presence of LVI (OR 8.64, 95% CI 7.32–10.24, *p* < 0.001), tumor size (OR 1.05, 95% CI 1.03–1.07, *p* < 0.001), and the number of lymph nodes sampled (OR 1.04, 95% CI 1.03–1.05, *p* < 0.001) (Table 3).

Furthermore, LN+ predictors were evaluated by multivariable logistic regression analysis separately for both black and white women (Table 4). The factors significantly associated with LN+ among black women included age (OR 0.98, 95% CI 0.96–0.998, *p* = 0.03), higher tumor stage (pT1b versus pT1a, OR 2.40, 95% CI 1.65–3.52, *p* < 0.001; pT2 versus pT1a, OR 4.99, 95% CI 3.31–7.54, *p* < 0.001), higher tumor grade (grade 2 versus grade 1, OR 2.40, 95% CI 1.46–4.09, *p* = 0.001; grade 3 versus grade 1, OR 4.01, 95% CI 2.46–6.78, *p* < 0.001), LVI (OR 5.48, 95% CI 3.95–7.65, *p* < 0.001), and tumor size (OR 1.06, 95% CI 1.006–1.12, *p* = 0.03). The factors that were significantly associated with LN+ among white women included age (OR 0.98, 95% CI 0.97–0.99, *p* < 0.001), insurance status (Medicaid versus uninsured, OR 1.45, 95% CI 1.07–1.97, *p* = 0.02), higher tumor stage (pT1b versus pT1a, OR 3.09, 95% CI 2.77–3.45, *p* < 0.001; pT2 versus pT1a, OR 5.07, 95% CI 4.42–5.80, *p* < 0.001), higher tumor grade (grade 2 versus grade 1, OR 1.41, 95% CI 1.27–1.57, *p* < 0.001; grade 3 versus grade 1, OR 1.37, 95% CI 1.21–1.56, *p* = 0.003), LVI (OR 6.56, 95% CI 5.96–7.22, *p* < 0.001), and tumor size (OR 1.05, 95% CI 1.04–1.06, *p* < 0.001). These results demonstrate that tumor grade was a stronger predictor of LN+ among black women while Medicaid insurance status predicted LN+ uniquely among white women.

Individual nomograms for predicting regional LN+ (Figure 2), pelvic only LN+ (Appendix A), and paraaortic +/− pelvic LN+ (Appendix A). Of note, the presence of LVI was a stronger predictor of paraaortic lymph node involvement relative to pelvic only LN+ (Table 3). Furthermore, patient race predicted paraaortic but not pelvic only LN+ while insurance status and academic facility type predicted pelvic LN+.

## 4. Discussion

Multiple studies have identified risk factors for lymph nodal involvement in endometrial cancer. GOG 33 strongly correlated risk of lymph node involvement with higher tumor grade and deeper myometrial invasion [12]. A subsequent study, GOG 210, confirmed the predictive value of primary tumor grade and stage in predicting lymph node involvement, as well as describing additional predictors of nodal positivity, including non-endometrioid histology and the presence of lymphovascular invasion [9].

Our analysis of over 35,000 patients was limited to those with endometrial adenocarcinoma and pathologic tumor stage I–II. We correlated risk of lymph node involvement with multiple tumor characteristics, including pathologic primary tumor stage, pathologic grade, tumor size, and LVI. We also identified novel correlations with demographic variables, including patient age, race, and insurance status. We also explored the relative capacity of these predictors to determine the risk of pelvic only versus paraaortic with or without pelvic lymph node involvement.

Although increasing patient age is an established predictor of clinical outcomes, including disease recurrence and survival in early-stage endometrial cancer [13,14], it is not clear whether patient age predicts regional lymph node involvement. The results of GOG 210 suggest that post-menopausal women are at a higher risk of both pelvic and paraaortic lymph node involvement than pre-menopausal women [9]. However, factors beyond patient age alone may impact the risk of nodal involvement when patients are stratified by menopausal status. In the present study, we found increasing patient age to be associated with a decreased risk of LN+ of approximately 2% per year. Multiple smaller retrospective studies have not found that age can predict risk of regional LN+ generally [15,16,17] or paraaortic LN+ specifically [18,19]. Another study of early-stage endometrial cancer patients with LVI found that node-negative patients were older [20]. A prior NCDB study found increasing age to be a predictor of lower risk of LN+ only among women with stage T1a disease [21]. To our knowledge, this is the first study to demonstrate a clear linear relationship between increasing patient age and lower risk of LN+.

We also found race to be a significant predictor of regional lymph node involvement with an associated increased risk of 19% in multivariable analysis, independent of pathologic or other sociodemographic factors. Interestingly, race was not a predictor of pelvic only LN+, while both black race and other race predicted a significantly higher risk of paraaortic LN+ of 64% and 54%, respectively, relative to white women. The limited prior studies have not reported a significant impact of patient race on the risk of LN+ in early-stage endometrial cancer [20,21]. The present study reports for the first time, to our knowledge, that black race is associated with regional LN+ generally and paraaortic LN+ specifically. Independent subset analyses of black and white women revealed that tumor grade was a stronger predictor of LN+ among black women, suggesting that more aggressive, higher grade tumor biology may put black women at a higher risk of developing lymph node metastasis. Although multiple sociodemographic factors were not significantly associated with the risk of LN+ in this study, non-biologic causes of the observed difference in LN+ risk by race cannot be readily excluded by this analysis. In fact, black women and other non-white women may be diagnosed with more clinically advanced disease for reasons independent of tumor biology that are not readily appreciated in the current analysis. Future studies may help to elucidate the underlying mechanisms of higher risk of lymph node positivity among black women as observed in this study.

The implications and underlying drivers of these observed differences remain unclear. Previous studies have demonstrated that black women have poorer survival relative to white women with endometrial cancer [22,23,24,25], even in stage for stage comparisons [22,23]. Young black women in particular have been found to have poorer survival compared to white women with endometrial cancer, and this survival disparity was found to be more prominent among women with early-stage disease. In the present study of women with early-stage endometrial cancer, black race and younger age were both found to be strong predictors of LN+, which may be a surrogate for more aggressive tumor biology that could drive differences in survival in these patient populations. However, the present study did not address how other patient sociodemographic factors or treatment may impact differences in earl-stage endometrial cancer patients stratified by race.

This study included nomograms to predict any regional LN+, pelvic LN+, and paraaortic +/− pelvic LN+. The presence of LVI, higher primary pathologic tumor stage, higher grade, larger tumor size, and younger patient age were all significant predictors included in each nomogram. Patient race predicted overall regional LN+ and paraaortic LN+ specifically but did not strongly predict pelvic LN+. Additionally, academic facility type predicted regional LN+ generally and pelvic LN+ while insurance status predicted pelvic LN+ only. Many studies created nomograms for LN+ risk [15,16,17,26,27], but the present study used, to our knowledge, the largest sample to create predictive nomograms including demographic factors for LN+ in uterine cancer patients. Furthermore, inclusion of sociodemographic variables beyond pathologic factors has been limited in previous nomograms. The nomograms in the present study have the advantage of incorporating patient age and race, both of which correlated with LN+.

The present study is limited by its retrospective nature and use of registry data with the possibility of unmeasured confounding variables not recorded in the NCDB, such as percentage depth of myometrial invasion and regional metastatic tumor deposit size, potentially impacting the reported results. The NCDB also does not account for variability in scoring systems utilized to define pathologic tumor grade or LVI between reporting institutions, thereby masking potential variability in reporting heterogeneity. It is also acknowledged that the number of lymph nodes sampled as a predictor of LN+ is inherently vulnerable to sampling bias, particularly given that the number of pelvic versus paraaortic lymph nodes sampled is not specified in the NCDB. Accordingly, the increasing rate of paraaortic lymph node positivity associated with increased number of lymph nodes sampled may be a function of the number of paraaortic lymph nodes sampled. Furthermore, there are limitations of the specificity of certain sociodemographic variables in the NCDB. For example, patient race is self-reported and susceptible to associated bias. Furthermore, demographic variables, including income and education, are reported based upon median values in the patient’s home address zip code rather than actual patient level data.

## 5. Conclusions

This study represents, to our knowledge, the largest study to date of sociodemographic and pathologic risk factors for lymph node involvement in patients with endometrial adenocarcinoma. Our results demonstrated associations between lymph node involvement and multiple established pathologic risk factors, including the presence of LVI, higher tumor stage, higher tumor grade, and larger tumor size in a nationally representative multi-institutional cohort. We also demonstrated, to our knowledge for the first time in the literature, that both younger age and black race are associated with a higher risk of LN+. Higher rates of lymph node positivity in these patient groups may be a surrogate for underlying aggressive tumor biology, which may contribute to higher mortality rates observed in these populations. Our findings help to confirm established and identify new predictors of lymph node involvement in clinically apparently node-negative endometrial adenocarcinoma and assess the composite impact of these variables on the risk of lymph node involvement by nodal drainage location.

## Figures and Tables

**Figure 2 biology-12-00982-f002:**
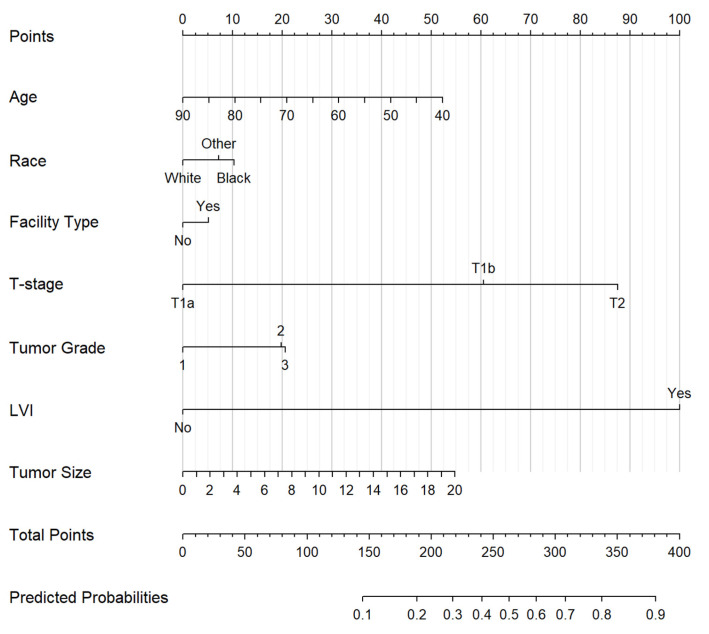
Nomogram for predicting risk of regional LN+ with tumor size in centimeters.

**Table 1 biology-12-00982-t001:** Distribution of demographic and pathologic tumor variables for the full cohort and by patient race. Multiple sociodemographic and pathologic factors varied significantly between patients of white versus black versus other race.

	Full Cohort	White	Black	Other Race	*p*
	N = 35,170	N = 31,274	N = 2329	N = 1567	
Age, years					
Median (IQR)	62.0 (56.0; 69.0)	62.0 (57.0; 69.0)	62.0 (56.0; 68.0)	59.0 (53.0; 66.0)	<0.001
Mean (SD)	62.7 (9.47)	63.0 (9.47)	62.0 (8.95)	59.4 (9.57)	<0.001
Charlson–Deyo Comorbidity Score					<0.001
0	26,430 (75.1%)	23,680 (75.7%)	1597 (68.8%)	1153 (73.6%)	
1	7168 (20.4%)	6262 (20%)	571 (24.5%)	335 (21.4%)	
2	1278 (3.6%)	1083 (3.5%)	130 (5.6%)	65 (4.2%)	
3	294 (0.8%)	249 (0.8%)	31 (1.3%)	14 (0.9%)	
Insurance Status					<0.001
Uninsured	1213 (3.5%)	1001 (3.2%)	123 (5.3%)	89 (5.7%)	
Private Insurance	18,339 (52.1%)	16,429 (52.5%)	1036 (44.5%)	874 (55.8%)	
Medicaid	1660 (4.7%)	1241 (4.0%)	225 (9.7%)	194 (12.4%)	
Medicare	13,513 (38.4%)	12,227 (39.1%)	923 (39.6%)	363 (23.2%)	
Other Insurance	445 (1.3%)	376 (1.2%)	22 (0.9%)	47 (3.0%)	
Income					<0.001
<$48,000	13,496 (38.4%)	11,630 (37.2%)	1450 (62.3%)	416 (26.5%)	
≥$48,000	21,674 (61.6%)	19,644 (62.8%)	879 (37.7%)	1151 (73.5%)	
Education					<0.001
Low	13,841 (39.4%)	11,568 (37.0%)	1576 (67.7%)	697 (44.5%)	
High	21,329 (60.6%)	19,706 (63%)	753 (32.3%)	870 (55.5%)	
Practice Type					<0.001
Non-Academic	27,557 (78.4%)	24,713 (79%)	1497 (64.3%)	1347 (86%)	
Academic	7613 (21.6%)	6561 (21%)	832 (35.7%)	220 (14%)	
Pathologic Tumor Stage					<0.001
1a	22,240 (63.2%)	19,618 (62.7%)	1566 (67.2%)	1056 (67.4%)	
1b	10,067 (28.6%)	9172 (29.3%)	511 (21.9%)	384 (24.5%)	
2	2863 (8.1%)	2484 (7.9%)	252 (10.8%)	127 (8.1%)	
Pathologic Nodal Stage					<0.001
0	32,306 (91.9%)	28,781 (92%)	2097 (90%)	1428 (91.1%)	
IIIC1	2027 (5.8%)	1797 (5.8%)	145 (6.2%)	85 (5.4%)	
IIIC2	837 (2.4%)	696 (2.2%)	87 (3.7%)	54 (3.5%)	
Pathologic Tumor Grade					<0.001
1	15,324 (43.6%)	13,834 (44.2%)	796 (34.2%)	694 (44.3%)	
2	14,011 (39.8%)	12,530 (40.1%)	890 (38.2%)	591 (37.7%)	
3	5835 (16.6%)	4910 (15.7%)	643 (27.64%)	282 (18%)	
LVI					0.88
Absent	28,125 (80%)	25,005 (80%)	1871 (80.3%)	1249 (79.7%)	
Present	7045 (20%)	6269 (20%)	458 (19.7%)	318 (20.3%)	
Tumor Size (cm)					
Mean (SD)	3.94 (2.86)	3.9 (2.87)	4.53 (2.7)	3.86 (2.64)	<0.001

**Table 2 biology-12-00982-t002:** Demographic and pathologic tumor variables predicting regional lymph node involvement.

Covariates	Univariate		Multivariable	
	OR (95% CI)	*p*	OR (95% CI)	*p*
Age	0.995 (0.991 to 0.999)	0.02	0.98 (0.98 to 0.99)	<0.001
Race				
White	1.000	-	1.000	-
Black	1.28 (1.11 to 1.45)	<0.001	1.19 (1.01 to 1.40)	0.04
Other	1.12 (0.94 to 1.34)	0.20	1.14 (0.93 to 1.38)	0.21
Charlson–Deyo Comorbidity Score				
0	1.000	-	1.000	-
1	0.94 (0.86 to 1.04)	0.24	0.99 (0.88 to 1.10)	0.97
2	1.05 (0.86 to 1.28)	0.60	1.04 (0.83 to 1.30)	0.72
3	1.27 (0.85 to 1.83)	0.21	1.28 (0.82 to 1.93)	0.25
Insurance Status				
Uninsured	1.000	-	1.000	-
Private Insurance	0.81 (0.67 to 0.99)	0.04	1.13 (0.91 to 1.42)	0.28
Medicaid	1.29 (1.01 to 1.64)	0.04	1.42 (1.09 to 1.88)	0.01
Medicare	0.83 (0.68 to 1.02)	0.07	1.19 (0.94 to 1.51)	0.16
Other Insurance	0.87 (0.58 to 1.26)	0.47	1.12 (0.72 to 1.71)	0.61
Income				
<$48,000	1.000	-	1.000	-
≥$48,000	0.92 (0.85 to 0.997)	0.04	0.94 (0.85 to 1.04)	0.23
Education				
Low	1.000	-	1.000	-
High	0.92 (0.85 to 0.99)	0.03	1.03 (0.93 to 1.14)	0.53
Practice Type				
Non-Academic	1.000	-	1.000	-
Academic	1.01 (0.92 to 1.11)	0.78	1.11 (0.995 to 1.22)	0.06
Year of Diagnosis	1.03 (1.007 to 1.05)	0.01	1.01 (0.99 to 1.04)	0.42
Pathologic Tumor Stage				
1a	1.000	-	1.000	-
1b	5.36 (4.88 to 5.90)	<0.001	3.08 (2.78 to 3.42)	<0.001
2	10.66 (9.52 to 11.94)	<0.001	5.10 (4.50 to 5.78)	<0.001
Pathologic Tumor Grade				
1	1.000	-	1.000	-
2	2.22 (2.02 to 2.44)	<0.001	1.45 (1.31 to 1.61)	<0.001
3	3.48 (3.13 to 3.86)	<0.001	1.47 (1.30 to 1.65)	<0.001
LVI				
Absent	1.000	-	1.000	-
Present	10.44 (9.61 to 11.35)	<0.001	6.44 (5.88 to 7.05)	<0.001
Tumor size (cm)	1.15 (1.14 to 1.17)	<0.001	1.05 (1.04 to 1.06)	<0.001

**Table 3 biology-12-00982-t003:** Demographic and pathologic tumor variables predicting pelvic LN+ only (left) versus paraaortic +/− pelvic LN+ (right).

Covariates	Pelvic		Paraaortic	
	OR (95% CI)	*p*	OR (95% CI)	*p*
Age	0.98 (0.98 to 0.99)	<0.001	0.97 (0.96 to 0.98)	<0.001
Race				
White	1.000	-	1.000	-
Black	1.05 (0.86 to 1.26)	0.65	1.64 (1.27 to 2.09)	<0.001
Other	0.98 (0.76 to 1.24)	0.85	1.54 (1.12 to 2.07)	0.006
Charlson–Deyo Comorbidity Score				
0	1.000	-	1.000	-
1	0.99 (0.88 to 1.12)	0.93	0.99 (0.82 to 1.18)	0.88
2	1.14 (0.89 to 1.44)	0.31	0.84 (0.54 to 1.26)	0.43
3	1.28 (0.77 to 2.03)	0.31	1.26 (0.58 to 2.43)	0.53
Insurance Status				
Uninsured	1.000	-	1.000	-
Private Insurance	1.29 (0.99 to 1.70)	0.06	0.88 (0.63 to 1.25)	0.45
Medicaid	1.62 (1.18 to 2.25)	0.003	1.10 (0.72 to 1.68)	0.66
Medicare	1.35 (1.02 to 1.81)	0.04	0.93 (0.65 to 1.35)	0.67
Other Insurance	1.37 (0.83 to 2.20)	0.21	0.78 (0.34 to 1.60)	0.52
Income				
<$48,000	1.000	-	1.000	-
≥$48,000	0.92 (0.82 to 1.04)	0.17	0.98 (0.82 to 1.16)	0.79
Education				
Low	1.000	-	1.000	-
High	1.06 (0.94 to 1.19)	0.34	0.98 (0.82 to 1.16)	0.8
Practice Type				
Non-Academic	1.000	-	1.000	-
Academic	1.13 (1.002 to 1.27)	0.04	1.01 (0.85 to 1.21)	0.88
Year of Diagnosis	1.01 (0.98 to 1.04)	0.51	1.01 (0.97 to 1.06)	0.61
Pathologic Tumor Stage				
1a	1.000	-	1.000	-
1b	2.91 (2.58 to 3.28)	<0.001	3.52 (2.91 to 4.27)	<0.001
2	5.00 (4.34 to 5.76)	<0.001	5.33 (4.28 to 6.65)	<0.001
Pathologic Tumor Grade				
1	1.000	-	1.000	-
2	1.47 (1.31 to 1.65)	<0.001	1.38 (1.15 to 1.66)	<0.001
3	1.45 (1.27 to 1.67)	<0.001	1.40 (1.14 to 1.72)	0.001
Lymphovascular Invasion				
Absent	1.000	-	1.000	-
Present	5.77 (5.20 to 6.39)	<0.001	8.64 (7.32 to 10.24)	<0.001
Tumor Size (cm)	1.04 (1.03 to 1.06)	<0.001	1.05 (1.03 to 1.07)	<0.001

**Table 4 biology-12-00982-t004:** Multivariable logistic regression analysis for predictors of regional lymph node involvement among black (left) versus white (right) women.

Covariates	Black		White	
	OR (95% CI)	*p*	OR (95% CI)	*p*
Age	0.98 (0.96 to 0.998)	0.03	0.98 (0.97 to 0.99)	<0.001
Charlson–Deyo Comorbidity Score				
0	1.000	-	1.000	-
1	1.25 (0.86 to 1.79)	0.23	0.96 (0.85 to 1.08)	0.47
2	0.65 (0.28 to 1.36)	0.28	1.07 (0.84 to 1.35)	0.57
3	0.96 (0.20 to 3.40)	0.95	1.26 (0.78 to 1.96)	0.32
Insurance Status				
Uninsured	1.000	-	1.000	-
Private Insurance	0.96 (0.49 to 1.96)	0.86	1.14 (0.90 to 1.47)	0.3
Medicaid	1.16 (0.53 to 2.62)	0.71	1.45 (1.07 to 1.97)	0.02
Medicare	0.98 (0.48 to 2.11)	0.96	1.21 (0.94 to 1.59)	0.15
Other Insurance	0.56 (0.07 to 2.83)	0.52	1.17 (0.73 to 1.86)	0.5
Income				
<$48,000	1.000	-	1.000	-
≥$48,000	0.83 (0.56 to 1.22)	0.34	0.94 (0.84 to 1.05)	0.25
Education				
Low	1.000	-	1.000	-
High	1.05 (0.70 to 1.57)	0.81	1.04 (0.93 to 1.15)	0.52
Practice Type				
Non-Academic	1.000	-	1.000	-
Academic	1.19 (0.86 to 1.65)	0.28	1.07 (0.96 to 1.20)	0.22
Year of Diagnosis	0.96 (0.49 to 1.96)	0.4	1.009 (0.98 to 1.04)	0.49
Pathologic Tumor Stage				
1a	1.000	-	1.000	-
1b	2.40 (1.65 to 3.52)	<0.001	3.09 (2.77 to 3.45)	<0.001
2	4.99 (3.31 to 7.54)	<0.001	5.07 (4.42 to 5.80)	<0.001
Pathologic Tumor Grade				
1	1.000	-	1.000	-
2	2.40 (1.46 to 4.09)	0.001	1.41 (1.27 to 1.57)	<0.001
3	4.01 (2.46 to 6.78)	<0.001	1.37 (1.21 to 1.56)	0.003
Lymphovascular Invasion				
Absent	1.000	-	1.000	-
Present	5.48 (3.95 to 7.65)	<0.001	6.56 (5.96 to 7.22)	<0.001
Tumor Size (cm)	1.07 (1.01 to 1.13)	0.01	1.05 (1.04 to 1.06)	<0.001

## Data Availability

Data from the National Cancer Database (NCDB) are available from the American College of Surgeons upon requests made by affiliate of member institutions. The NCDB data use agreement prohibits the authors from sharing the data.

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
