# Peer review of "Demographic Factors Predict Risk of Lymph Node Involvement in Patients with Endometrial Adenocarcinoma"

_biology, 2023, doi:10.3390/biology12070982_

Round 1

Reviewer 1 Report

Very good well written article.

Author Response

Thank you

Reviewer 2 Report

In my opinion, the analyzed topic is interesting enough to attract the readers’ attention. The goal of this article was to explore demographic factors to predict the risk of lymph node involvement in patients with endometrial adenocarcinoma. I think that the abstract of this article is not clear and well structured. In fact, it should be reorganized in order to highlight the aim of the study. In my opinion, the discussion could be studied in depth and extended. Maybe, it could be useful the evaluation of the state of the art of new protocols of management of endometrial cancer starting from the molecular point of view. In particular I suggest these two articles  PMID: 36979434 and PMID: 36833105 in order to analyze in depth the topic. Because of these reasons, the article should be revised and completed. Tables are clear. Considered all these points, I think it could be of interest for the readers and, in my opinion, it deserves the priority to be published after minor revisions

moderate review of English grammar and language should be performed.

Author Response

Thank you for thoughtfully reviewing our manuscript. Unfortunately, pertinent molecular information was not available for analysis in the NCDB dataset.

Reviewer 3 Report

This retrospective study evaluated the risk factors for lymph node positivity in a large cohort of endometrial cancer patients. The risk factors for lymph node positivity are well known and have been reported in numerous studies. The authors only briefly mention the role of sentinel node biopsy and do not mention whether pelvic and paraaortic lymphadenectomy is necessary in all patients. In addition, the authors do not explain why only patients with at least 10 removed lymph nodes were analysed. Why did the authors include patients only up to 2016? The survival analysis according to risk factors is not performed and whole paragraphs in the discussion section are written without proper referencing.

Author Response

Thank you for thoughtfully reviewing our manuscript. The cutoff for inclusion was >0 LN sampled as indicated in the methods section of the manuscript, and the CONSORT diagram has been updated to reflect this. This study only included patients up to 2016 as these were the data available at the time of initial analysis. Survival analysis was not included in the results as this was not an objective of the study.

Reviewer 4 Report

Thank You for Your excellent work regrading prediction of lymph node involvement in patients with endometrial cancer.

I have some queries for authors:

1. Please add information how tumor size was assessed: preoperatively by imaging techniques (CT, MRI, US?) or postoperatively by pathologist?

2. Please explain what is the aim of generating nomograms regarding lymph node involvement which contain specific characteristics that can only be acquired postoperatively?  In my opinion, such nomograms are needed when planning surgery (plan lymphadenectomy/sentinel lymph node biopsy or not) and should include only preoperatively-available characteristics. In Your nomograms, LVI status, and pathologic stage was included, which can be obtained only from postoperative final pathology report. 

3. Please check the manuscript for some minor mistakes, i.e. line 208 - with or without; line 278 - correlated risk of

Author Response

Thank you for your thoughtful review of our manuscript.

Tumor size was assessed postoperatively and defined by the evaluating pathologist. This clarification was added to the methods section of the manuscript (Statistical Analysis subsection).

Knowledge of the likelihood of lymph node involvement as assessed by associated nomograms can be used to guide adjuvant treatment decisions in patients for which lymph node assessment was not performed.

Additional errors were revised accordingly.

Round 2

Reviewer 3 Report

I have no further comments